# Chiral acidic amino acids induce chiral hierarchical structure in calcium carbonate

Wenge Jiang[1], Michael S. Pacella[2], Dimitra Athanasiadou[1], Valentin Nelea[1], Hojatollah Vali[3], Robert M. Hazen[4], Jeffrey J. Gray[5] & Marc D. McKee[1,3]

Chirality is ubiquitous in biology, including in biomineralization, where it is found in many hardened structures of invertebrate marine and terrestrial organisms (for example, spiralling gastropod shells). Here we show that chiral, hierarchically organized architectures for calcium carbonate (vaterite) can be controlled simply by adding chiral acidic amino acids (Asp and Glu). Chiral, vaterite toroidal suprastructure having a 'right-handed' (counterclockwise) spiralling morphology is induced by L-enantiomers of Asp and Glu, whereas 'left-handed' (clockwise) morphology is induced by D-enantiomers, and sequentially switching between amino-acid enantiomers causes a switch in chirality. Nanoparticle tilting after binding of chiral amino acids is proposed as a chiral growth mechanism, where a 'mother' subunit nanoparticle spawns a slightly tilted, consequential 'daughter' nanoparticle, which by amplification over various length scales creates oriented mineral platelets and chiral vaterite suprastructures. These findings suggest a molecular mechanism for how biomineralization-related enantiomers might exert hierarchical control to form extended chiral suprastructures.

[1] Faculty of Dentistry, McGill University, Montreal, Quebec, Canada H3A 0C7. [2] Department of Biomedical Engineering, Johns Hopkins University, Baltimore, Maryland 21218, USA. [3] Department of Anatomy and Cell Biology, Faculty of Medicine, McGill University, Montreal, Quebec, Canada H3A 0C7. [4] Geophysical Laboratory, Carnegie Institution of Washington, 5251 Broad Branch Road NW, Washington, District of Columbia 20015, USA. [5] Department of Chemical and Biomolecular Engineering, Johns Hopkins University, Baltimore, Maryland 21218, USA. Correspondence and requests for materials should be addressed to M.D.M. (email: marc.mckee@mcgill.ca).

Chirality—the fundamental phenomenon of handedness—ranges across nature from the atomic arrangement of amino acids to the long, macroscopically helical tooth of the narwhal *Monodon monoceros*[1]. Chirality is ubiquitous in biology, including in calcium carbonate biomineralization, and it is found in many hardened structures of invertebrate marine and terrestrial organisms, notably helical gastropod shells and now-extinct ammonites[2–6]. In nonbiological systems, formation of chiral crystals has been described, induced by abiotic twist, optical effects and surface stress[7–9]. In biology, differences in the handedness of biomineralized, chiral architectures of calcium carbonate polymorphs are thought to result from the actions of chiral biomolecules[4,10–15]. However, until now, knowledge of how chiral molecules might direct nano-sized calcium carbonate 'building blocks' to form larger chiral hierarchical architectures remains unknown. Vaterite, a polymorph of $CaCO_3$, has attracted attention in the biomineralization field because it is the mineral phase of the solitary stolidobranch ascidian *Herdmaniamomus*[16], an invertebrate having a hierarchically organized helical skeleton. In addition, vaterite is the mineral phase produced during gastropod helical shell repair and in otoliths of marine fishes[17,18]. In the inner ear of humans, otoconia of the vestibular apparatus—which functions in balance by sensing gravity and linear acceleration—can show abnormal, pathologic rounded chiral vaterite structures rather than the appropriate calcitic morphology having well-defined crystalline faces as found in healthy individuals[3]. A remarkable example of change in chirality can be seen in the helical shell of the marine foraminifer *Globigerina pachyderma*, where in the Arctic and Antarctic oceans the mineralized $CaCO_3$ shells grow in a right-handed (dextral, counterclockwise) direction; however, for unknown reasons, in temperate and tropical waters, left-handed (sinistral, clockwise) shells predominate[4,5].

Biomineralization processes occur within organisms of most phyla and have as a hallmark feature the regulation of crystal growth by proteins rich in acidic amino-acid residues[19]. Of particular interest, these acidic amino acids—Asp and Glu—have been implicated in biochemical homochirality[20,21]. To determine whether such chiral acidic amino acids can transmit chiral information to hierarchical structures of biologically influenced minerals (that is, biominerals), we investigated the effects of chiral Asp and Glu enantiomers on vaterite deposition and growth.

Here we show that complex, chiral hierarchically organized architectures for the calcium carbonate mineral vaterite can be induced simply by chiral Asp and Glu enantiomers via a nanoparticle tilting growth mechanism. In this process, chiral vaterite toroidal suprastructure having 'right-handed' spiralling morphology is induced by the addition of L-enantiomers of Asp and Glu, whereas 'left-handed' spiralling morphology is induced by D-enantiomers.

## Results

### Chiral vaterite toroids induced by chiral acidic amino acids.
Scanning electron microscopy (SEM) of calcium carbonate crystals grown in supersaturated calcium- and carbonate-containing solution in the absence of amino acids revealed predominant characteristic rhombohedral calcite crystals (the most thermodynamically stable phase of $CaCO_3$ at ambient conditions), and occasional vaterite crystals that were achiral and had hexagonal symmetry (Fig. 1a,b). However, calcium carbonate crystals grown in the presence of the chiral acidic amino acids Asp and Glu showed that the predominant mineral phase was vaterite, which formed complex structures that had chiral features (Figs 1c,d and 2). Similar to our work,

which produced a mix of polymorphs (both calcite and vaterite) after adding amino acid (but, where vaterite predominated), other studies using different solution conditions have also reported the formation of a mix of polymorphs in the presence of amino acids[22]. In the present study, vaterite formed hierarchically organized, toroid-shaped chiral suprastructures where the spirally oriented morphology of assembled mineral platelets—somewhat like the blades of a propeller or a pinwheel—depended upon the enantiomeric form of the acidic amino-acid additive (Figs 1 and 2 and Supplementary Fig. 1).

In the presence of D- and L-enantiomers of the acidic amino acids Asp and Glu, hierarchical organization of aligned, flat crystal platelets with rounded/curved edges formed to produce spiralling toroidal suprastructures—chiral (vaterite) toroids—that were mirror images of each other. In the presence of Asp enantiomers, a spiral and aligned array of crystal platelets formed as rounded vaterite toroids, where the chiral directions of these spirally oriented crystal platelets could be characterized as being either counterclockwise (right-handed) or clockwise (left-handed) according to whether L- or D-Asp was added, respectively (Fig. 2a,b). Mineral growth under racemic (50:50) L- and D-Asp conditions resulted in symmetric vaterite with no chiral character (Fig. 2c).

When Glu enantiomers were used in a manner similar to Asp enantiomers, the spirally arranged and aligned vaterite platelets likewise assembled into spiralling suprastructures having a counterclockwise sense for L-Glu and a clockwise sense for D-Glu (Fig. 2d,e). The addition of Glu, however, resulted in toroidal morphology with a central core area not showing any platelets, as well as a substructure within the chiral vaterite toroids consisting of six spirally oriented platelet domains (Fig. 2d,e). When a racemic Glu mixture was added to the growth solution, no chiral structuring was observed, and a symmetric nontoroidal hexagonal form of vaterite was formed (Fig. 2f). The chirality of vaterite toroids did not depend on the pH of the growth solution, and the chiral morphologies formed in a broad range of pH (from slightly acidic at pH 6.46 to basic at pH 12.40; Fig. 3). As additional controls, when enantiomers of neutral and basic chiral amino acids such as Ala and Lys (respectively) were used, or the simplest achiral amino acid Gly was added, the vaterite forms were symmetrically hexagonal, nontoroidal and achiral (Fig. 4).

The precipitation of $CaCO_3$ in the presence of amino acids, including acidic amino acids, has been widely studied because of its importance in various areas of crystallization, biomineralization and geology[22–27]. Indeed, in terms of controlling calcium carbonate polymorph formation, work similar to ours has shown induction of symmetric/achiral vaterite by acidic amino acids[23–26]. However, we show here for the first time that chiral, hierarchical vaterite toroidal suprastructures can be induced by chiral acidic amino acids. We believe that this novel observation derived from our using relatively low concentrations of calcium and carbonate ions (low supersaturation level), and longer growth times, as compared to the previous studies. To grow calcium carbonate mineral, two main methods are generally used: (i) the fast (minutes to hours) ammonia-diffusion method that quickly results in a high carbonate ion concentration and a high pH solution (from basic ammonia ions) and (ii) the high concentration method of solution $CaCl_2$ and $Na_2CO_3/NaHCO_3$, both of which result in a very high supersaturation state for vaterite (at least 100 times greater than that used in our method)[22–26]. These previous, high-mineral ion concentration studies resulted in faster precipitation of vaterite (minutes to hours) compared to our slower (hours to days) process. Consequently, under the fast-growth conditions used by others, the high concentration of calcium and carbonate ions dominated

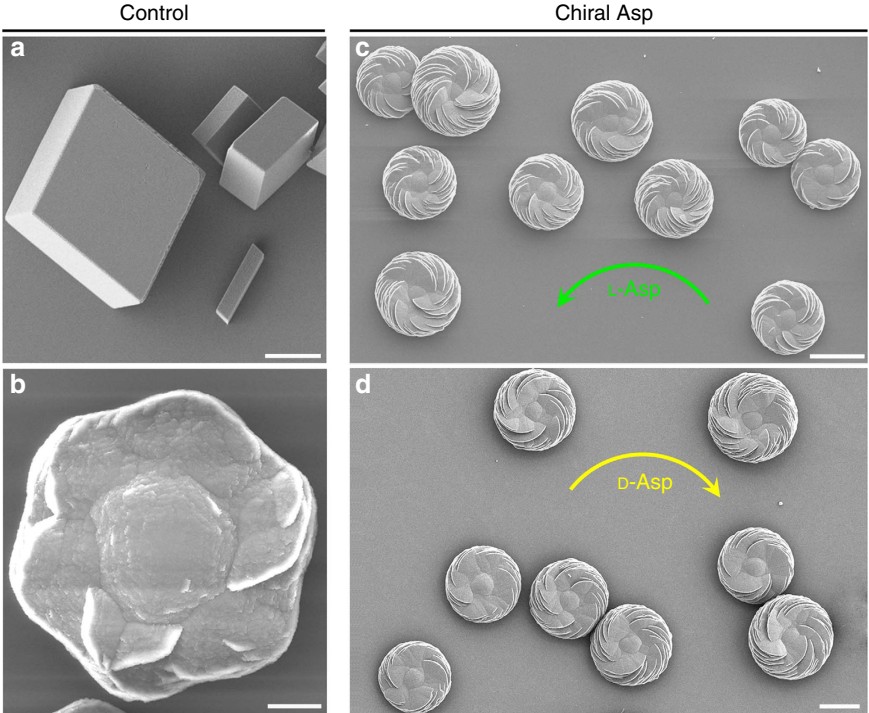

**Figure 1 | Uniformity of chiral vaterite toroids induced by chiral acidic amino acids.** Whereas characteristic rhombohedral calcite crystal morphology (**a**) and hexagonal vaterite morphology (**b**), as viewed by SEM, both form in the absence of amino acids when grown in a supersaturated calcium carbonate solution, the vaterite mineral phase predominates when grown in 20 mM L- or D-Asp (**c,d**), and all chiral toroids spiral in the counterclockwise direction (green arrow, **c**) or clockwise direction (yellow arrow, **d**) for L- and D-Asp, respectively. All chiral vaterite toroids have uniform and reproducible morphology. Scale bars, 8 μm (**a,b**) and 15 μm (**c,d**).

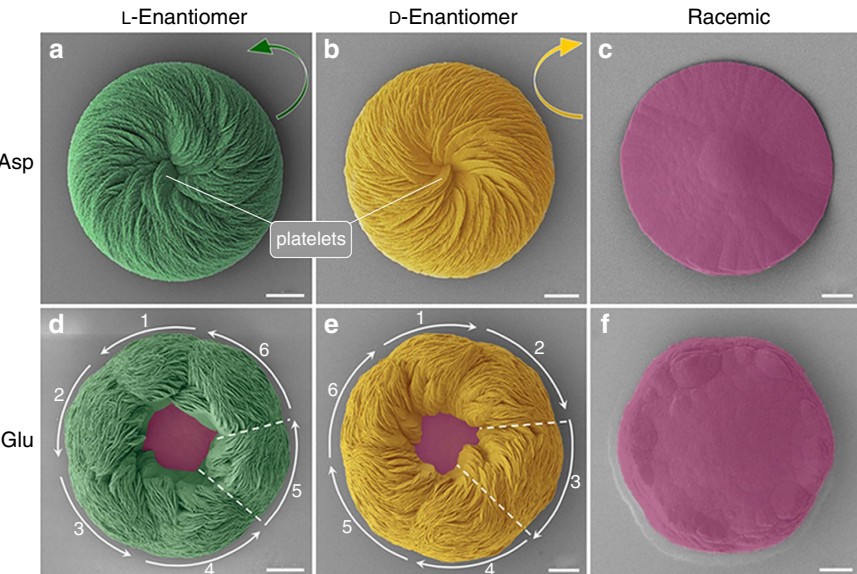

**Figure 2 | Hierarchical vaterite toroid suprastructures showing chiral orientations.** SEM images of vaterite toroids grown in 20 mM L- or D-Asp or in a racemic mixture of Asp (**a–c**), or in 20 mM L- or D-Glu or in a racemic mixture of Glu (**d–f**; pseudocoloured). L-enantiomers produce chiral toroids having a counterclockwise (right-handed) spiralling morphology (green arrow, **a,d**), whereas D-enantiomers produce a clockwise (left-handed) spiralling morphology (yellow arrow, **b,e**). For both L- and D-Glu, toroids form with a central core region remaining uncovered by platelets and with six peripheral structural subdomains, as indicated by the numbered white curved arrows and dashed lines (**d**). Under racemic conditions, no chirality effect is observed (**c,f**). Vaterite platelets are coloured green and yellow as induced by L- and D-acidic amino acids, respectively, and achiral vaterite structure is coloured pink. Scale bars, 6 μm (**a,b,f**) and 8 μm (**c–e**).

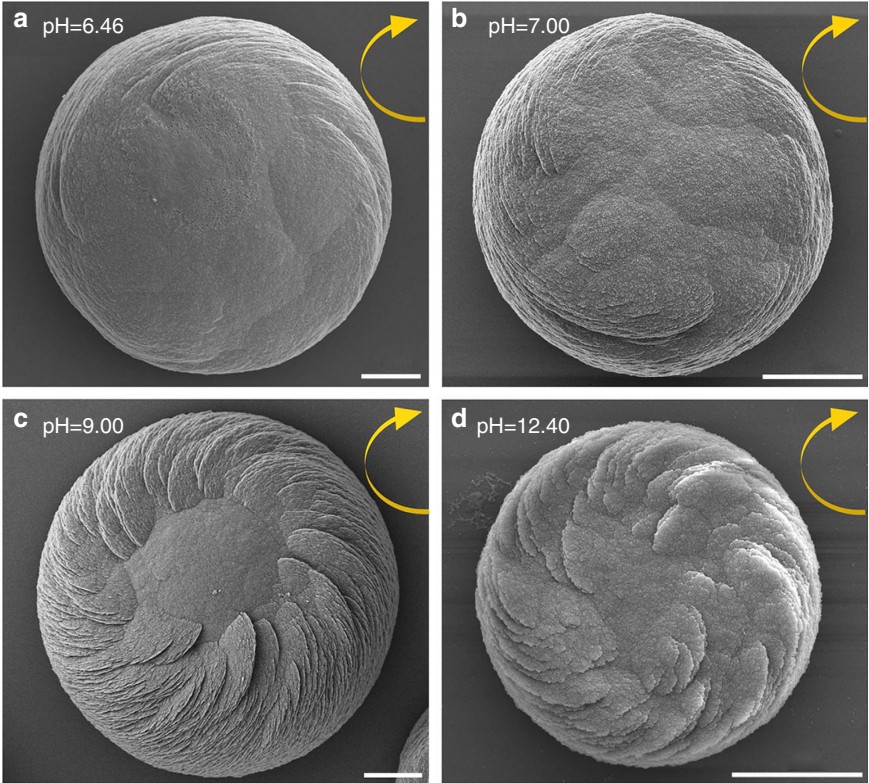

**Figure 3 | Chiral vaterite toroid formation at different pH.** SEM images of vaterite toroids formed in the presence of D-Asp in different pH solutions (**a–d**; pH = 6.46 for 10 d, pH = 7.00 for 7 d, pH = 9.00 for 3 d and pH = 12.40 for 6 h, respectively), with all showing the same clockwise spiral growth direction (yellow arrows). Scale bars, 6 μm (**a–c**) and 2 μm (**d**).

the dynamics of vaterite growth and symmetric vaterite structure was formed, as contrasted to the spiralling chiral effects we observed for vaterite by presenting acidic amino-acid enantiomers under slower calcium carbonate growth conditions (Figs 1–3 and Supplementary Fig. 1).

**Growth evolution of vaterite into chiral hierarchical toroids.** Normally, in the absence of additives, vaterite crystals belong to the symmetric hexagonal crystal space group and thus show no chiral features[16,28]. To understand how such complex vaterite structures develop over time with the addition of acidic amino acids, SEM and atomic force microscopy (AFM) were used to examine the time-course evolution of the chiral toroids (Fig. 5 and Supplementary Fig. 2). An example is illustrated using L-Asp in the growth solution (Fig. 5a–i). Initially, a round flat disc was formed (Fig. 5a) that consisted of $CaCO_3$ nanoparticles roughly 20 nm in size as measured by SEM and AFM (Fig. 5b). These flat $CaCO_3$ discs were amorphous as demonstrated by micro-Raman spectroscopy (Fig. 6) and X-ray diffraction (Supplementary Fig. 3), and they showed no chiral features. However, with time, the amorphous $CaCO_3$ transformed into crystalline vaterite[26,29,30], and, perhaps most importantly in terms of the early events of induction of chirality, some spiralling, short vaterite platelets emerged (white arrows in Figs 5c and 6) with a counterclockwise growth direction from the outer-edge region of the initial disc, as arising from the effect of the added L-Asp enantiomer. With additional time (Fig. 5d,e), continued formation and growth of spirally oriented vaterite platelets occurred predominately at the outer-edge region of the discs. Additional platelets with the same growth direction then formed on top of previous platelets to increase the overall

thickness of the initial flat discs to form larger structures that we refer to here as vaterite toroids. Such a growth pattern decreased the dimensions of the achiral central core region of the initial disc (Fig. 5c–f). With time, platelets converged on the central core region, completely covering the whole initial disc and resulting in thick toroidal suprastructures of vaterite having a spiral, counterclockwise morphology. All platelets possessed a nanoparticle substructure that was exclusively vaterite, with no amorphous phase being detectable in the platelets throughout the whole growth process (Figs 5g,h and 6 and Supplementary Movies 1 and 2). Similar to the results obtained using L-Asp, use of D-Asp reproduced all formation and growth processes of vaterite to produce spiralling, chiral toroids having an opposite, clockwise growth pattern (Supplementary Fig. 2 and Supplementary Movie 1).

For the Glu enantiomer effects on vaterite growth, the formation of chiral toroids was essentially similar but with differences at the intermediate and latest stages of growth. Using L-Glu as an example, the round, flattened initial discs likewise were composed of amorphous $CaCO_3$ nanoparticles as the first step in their formation (Fig. 5j,k), with oriented vaterite platelets likewise forming at the outer-edge region of the initial disc (Fig. 5l,m). However, growth evolution over time resulted in toroidal morphology, as well as the appearance of six distinct interlacing domains of platelet organization contributing to the larger chiral suprastructure (Fig. 5n–p), leaving the central core region exposed without being covered by platelets of vaterite (Fig. 5p), even after 3 weeks of growth. Similar to the effects of added L-Glu, growth in D-Glu produced identical chiral toroids but having an opposite, clockwise morphology (Supplementary Fig. 2). This biomineral chirality induced by chiral acidic amino acids causing rounding of vaterite platelet

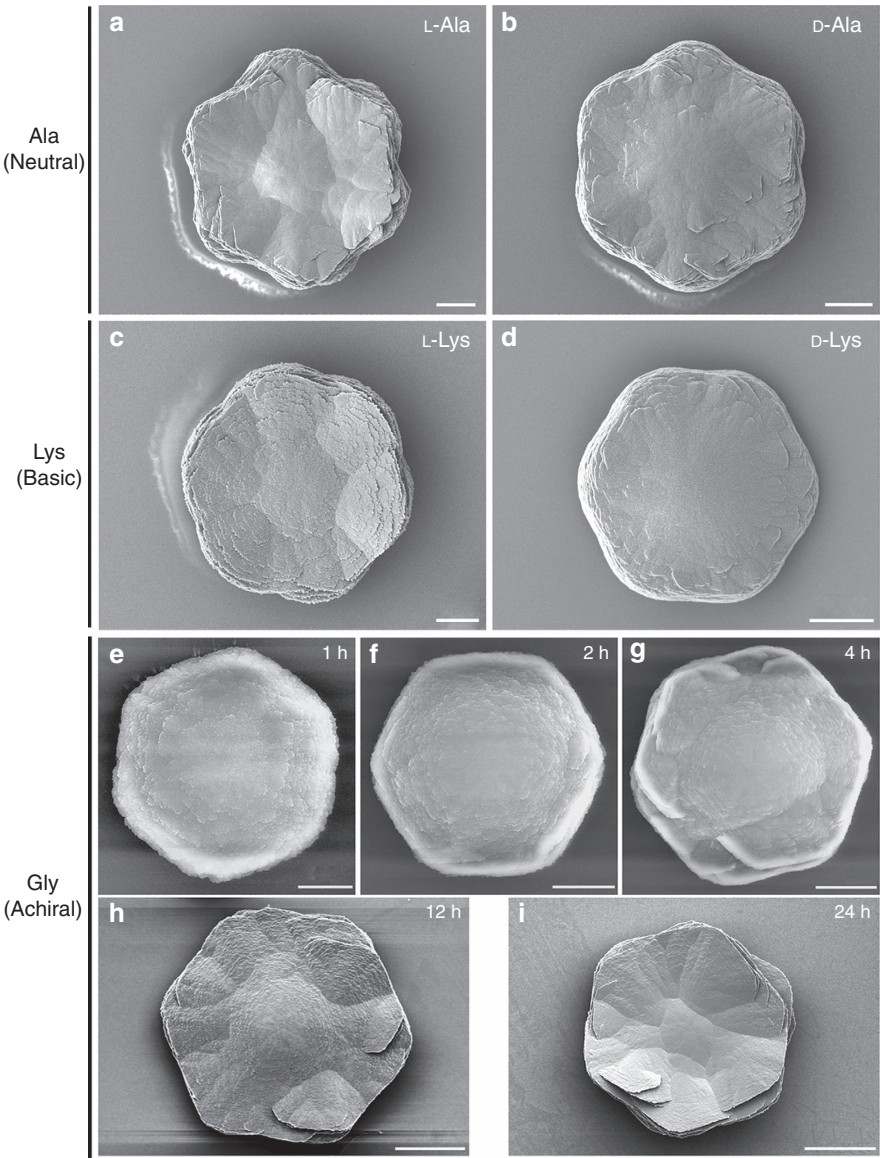

**Figure 4 | Symmetric vaterite grown in nonacidic amino acids.** Unlike the chiral structures formed using chiral acidic amino acids, characteristic hexagonal symmetric vaterite crystals form in the presence of neutral (**a**) L-alanine (L-Ala) and (**b**) D-Ala, and basic (**c**) L-lysine (L-Lys) and (**d**) D-Lys for 8 h, and achiral glycine (Gly) after 1 h (**e**), 2 h (**f**), 4 h (**g**), 12 h (**h**) and 24 h (**i**) of growth. Scale bars, 10 µm (**a–d**), 2 µm (**e**), 4 µm (**f**), 8 µm (**g**), 12 µm (**h**) and 20 µm (**i**).

edges contrasted sharply with the symmetric hexagonal growth that occurred in the absence of amino acids, where hexagonally shaped vaterite platelets formed with development (Fig. 1b and Supplementary Fig. 4).

**Chiral switching of toroids by changing amino-acid enantiomer.** Given our demonstration of biomineral chirality induced by chiral acidic amino acids, we next examined whether switching between the L- and D-enantiomeric forms of Asp and Glu could switch the respective chirality of the toroids. Indeed, abrupt chiral transitions in suprastructure architecture were obtained by simply replacing one enantiomer with the other, here shown for Glu. Initially, L-Glu imparted a counter-clockwise morphology to the vaterite toroid (green arrow, Fig. 7a). Nascent small platelets (black arrows) branched off from existing larger platelets (asterisks) and had a spiral counter-clockwise growth direction (long green arrows) that was switched

to the spiral clockwise growth direction (long yellow arrows) when the initial L-Glu-containing solution was replaced by a D-Glu-containing solution (Fig. 7b–f). As part of this switch in orientation, initial spiralling platelets having a counterclockwise growth direction disappeared and transitional symmetric platelets were formed (Fig. 4b), with subsequent spiral organization of the platelets in exclusively the clockwise sense starting at 2 days after the switch in solutions (yellow arrows, Fig. 7c–f). With the continued addition and growth over time of more spiralling platelets with clockwise growth direction, all transitional symmetric platelet surfaces were buried within the toroid and the chiral switch completed to end with the six outersurface structural subdomains as seen previously when only one Glu enantiomer was used (Fig. 2e). As observed for D-Glu alone, switching the enantiomeric form of L- to D-Glu caused with time the partial closure of the central achiral core region of the toroid (Fig. 7d,e). In addition, we compared the growth direction of the vaterite platelets on mature vaterite

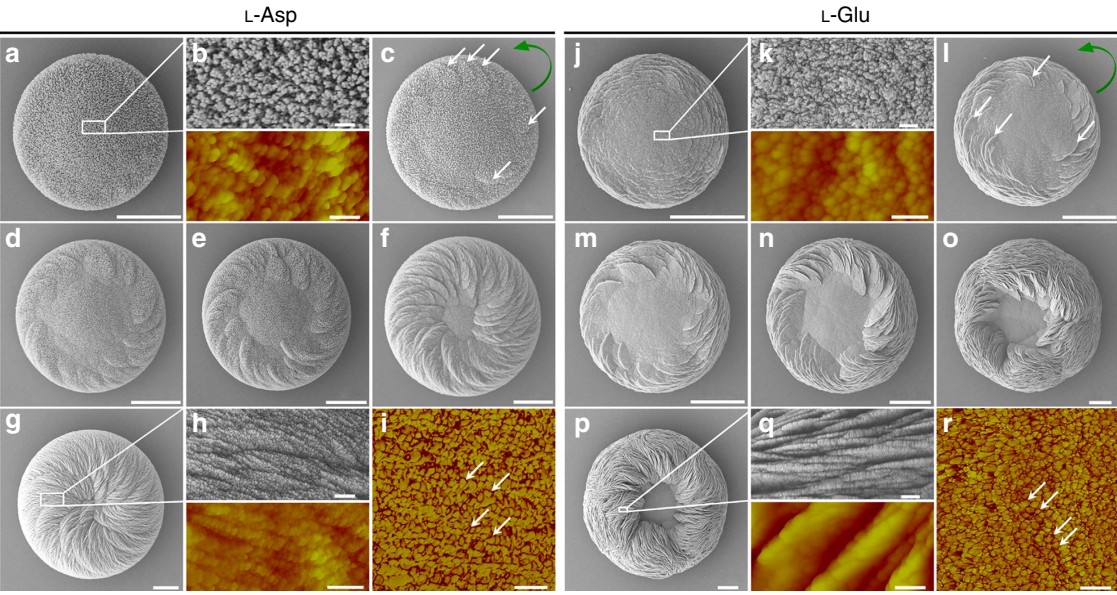

**Figure 5 | Growth evolution of chiral vaterite toroids grown in L-Asp or L-Glu.** SEM and AFM (coloured) images of toroid growth for 2 h (**a,b**), 3 h (**b**), 5 h (**d**), 8 h (**e**), 12 h (**f**) and 24 h (**g,h**) in the presence of L-Asp, and for 1.5 h (**j,k**), 2 h (**l**), 3 h (**m**), 5 h (**n**), 10 h (**o**) and 24 h (**p,q**) in the presence of L-Glu. Oriented vaterite platelets (white arrows) emerge within several hours (**c,l**) at the outer edge of a flat substrate vaterite disc to begin the formation of a toroid with a counterclockwise chirality (green curved arrow), and these and other platelets continue spiral growth to encroach upon, and then obscure (in the case of Asp), the centrally located achiral vaterite core region. SEM images (upper panels of **b,h,k,q**) and AFM height images (lower panels of **b,h,k,q**) show that all vaterite elements in the toroids have nanoparticle substructure. AFM phase mode (**i,r**) shows inorganic vaterite nanoparticles (yellow) surrounded by organic amino acid (red) in the platelets. Scale bars, 4 µm (**a,c–g**), 200 nm (**b,i,k,r**), 400 nm (**h**), 10 µm (**j,l–p**), 400 nm (upper panel of **q**) and 200 nm (lower panel of **q**).

toroids formed in pure L-Glu solution with that which occurred after replacing L-Glu with D-Glu, and they indeed had opposite growth directions (Fig. 7g,h). High-magnification SEM images showed chiral branching growth directions of nascent small vaterite platelets (black arrows) arising from larger existing platelets (asterisks) that were counterclockwise for pure L-Glu, but changed to the clockwise direction after replacing L-Glu with D-Glu (Fig. 7g,h). Thus, without any change in temperature, pH or stirring, the switching of chiral enantiomers of acidic amino acids in solution was sufficient to induce in the vaterite a chiral switch in the growth pattern of the toroids.

**Tilting of vaterite nanohexagons induces curved-edge platelets.** Similar to ultrastructural findings by Hu *et al.*[26], high-resolution observations by AFM, SEM and transmission electron microscopy (TEM) demonstrated a nanoparticle substructure in both the symmetric platelets of pure (no additive) hexagonal vaterite, and in the spirally oriented curved platelets formed in the presence of Asp and Glu. Both symmetric hexagonal vaterite and chiral vaterite toroids were constructed from nanoparticle subunits aligned by continuous crystallization of nanoparticles (nanohexagonal prisms) at the platelet edge—whether by nucleation, through a growth instability or by attachment of unresolvable clusters (Figs 5 and 8 and Supplementary Figs 2, 4 and 5). No evidence was found for self-assembly of independent, initially dispersed single vaterite nanoparticles, this being confirmed by an absence of pre-existing nanoparticles in the reaction solutions.

In the absence of chiral acidic amino acids, vaterite platelets had a well-known hexagonal growth morphology as exemplified by straight edges and hexagonal angles over various length scales, resulting from perfect alignment of nascent

vaterite nanoparticles onto existing nanoparticles (Fig. 8a–c and Supplementary Fig. 4). In the presence of chiral acidic amino acids, as evidence that the chiral acidic amino acids were extensively incorporated into the oriented platelets similar to the incorporation of amino acids in calcite found by Pokroy *et al.*[31], we observed that the C–H peak of L-Asp seen by micro-Raman spectroscopy deviated by $\sim 19\,\mathrm{cm}^{-1}$ compared to that of pure L-Asp solid amino-acid powder (Supplementary Fig. 6). The amount of L-Asp incorporation was established biochemically to be $\sim 1.4\%$ by weight, this value being similar to that found in natural biomineralized structures—for example, in the mollusk shell—where the amount of organic matrix is 1–5% (ref. 32). This observation was also confirmed using the phase function of AFM in the tapping mode, which is extremely sensitive to surface inhomogeneity and variations in composition, and which is widely used to examine complex biominerals to distinguish between organic matrix and inorganic crystalline phases[33]. Here we show (Fig. 5i,r and Supplementary Figs 2 and 7) such heterogeneous subunit variation within the platelets, where inorganic crystalline vaterite nanoparticles (yellow) are separated by boundaries marking the position of organic material (red, acidic amino acids). On the basis of these findings, we hypothesize that the presence of enantiomeric amino acid at the surface of the nanoparticles causes a change in their alignment—a tilt between mother and daughter nanohexagonal prisms—that might generate platelet-edge curvature (Figs 1–8 and Supplementary Figs 1, 2 and 5), thus deviating from the otherwise precise co-alignment that occurs in the absence of amino acid resulting in straight edges and sharp angles (Fig. 8a–c and Supplementary Fig. 4). Such a tilt—in this case being $\sim 4°$—was documented between nanoparticles by high-resolution TEM where lattice plane visualization allowed angle measurements to be made after toroid growth in the presence of chiral amino acid (Fig. 8h and Supplementary

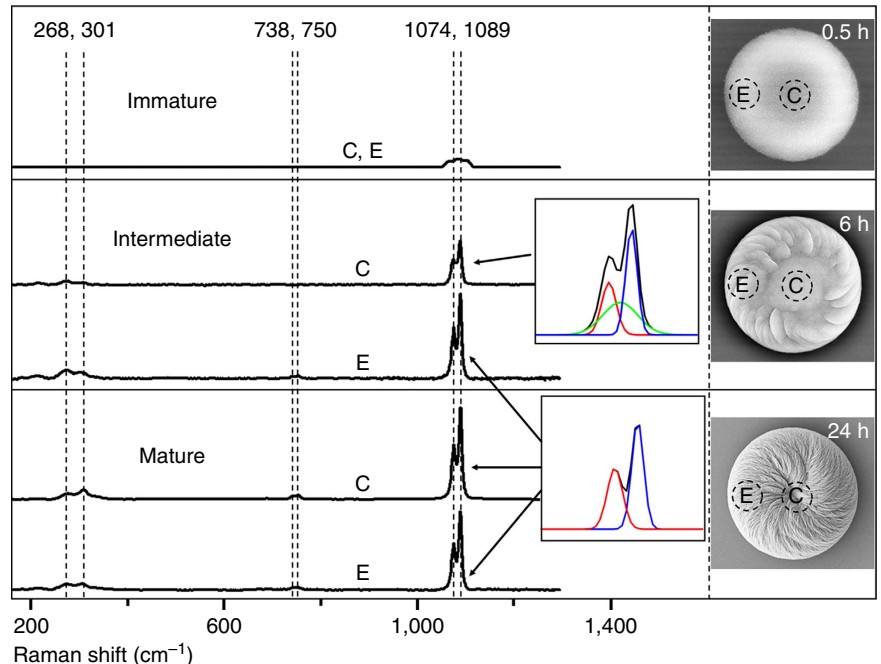

**Figure 6 | Mineral-phase evolution of chiral toroids grown in L-Asp.** Micro-Raman spectroscopy of early-stage calcium carbonate discs show only a single broad peak at ~1,085 cm$^{-1}$ (indicative of amorphous calcium carbonate) for both core (C) and edge (E) regions of a single immature initial disc without platelets, as observed by SEM. At the intermediate stage, where platelets are clearly present at the edge of the toroids as observed by SEM, the calcium carbonate $v_1$ peak of the spectra from the core region of an intermediate single toroid could be deconvoluted into one amorphous broad peak at ~1,085 cm$^{-1}$ (green peak) and two sharp vaterite peaks at 1,074 cm$^{-1}$ (red peak) and 1,089 cm$^{-1}$ (blue peak); these data indicate a phase transformation from the amorphous state to vaterite. This vaterite contribution in the core region is not high, since the $v_4$ peaks at 738 and 750 cm$^{-1}$ are not present. Micro-Raman analysis from the edge region of the intermediate toroids, where platelets are abundant, show a calcium carbonate $v_1$ peak composed only of the two sharp vaterite peaks at 1,074 and 1,089 cm$^{-1}$ without the amorphous broad peak; in addition, other vaterite $v_4$ peaks appear at 738 and 750 cm$^{-1}$, and lattice mode peaks at 268 and 301 cm$^{-1}$. These data indicate that the vaterite platelets form directly without a precursor amorphous phase. At the mature stage, here shown for L-Asp, where platelets are present at the entire surface of the toroid, all calcium carbonate peaks of the micro-Raman spectra—from both core and edge regions of a single mature toroid—can be assigned exclusively to vaterite, with no amorphous broad peaks being present.

Fig. 8). This slight offset between the 'mother' nanohexagon 1 and the consequential 'daughter' nanohexagon 2 was also confirmed by nanobeam electron diffraction in TEM to characterize the relationship between two adjacent nanohexagons (Fig. 8h).

**Mechanisms inducing toroid chirality by chiral amino acids.** Chiral biomineral growth driven by chiral amino acids is attributable to the interaction between chiral amino acids and specific mineral planes[20,34]. Using high-resolution TEM for lattice imaging, and focused-ion beam (FIB) cutting and thinning of the toroids followed by TEM (Supplementary Fig. 9) and X-ray diffraction (Supplementary Fig. 1), the hexagonal structure of the vaterite nanoparticles in the platelets was clearly identified. Here we visualized the basal (001) face and hexagonal boundaries, observations in agreement with previous reports for hexagonal vaterite (Fig. 9a)[26,29,30]. High-resolution AFM phase data obtained from the platelets showed that chiral acidic amino acids reside at the boundaries between vaterite subunit nanoparticles, implying that chiral acidic amino acids bind to the boundary plane, rather than to the (001) basal face (Fig. 5i,r, Supplementary Figs 2 and 7 and see Supplementary Discussion).

In order to explore the atomic-level interactions between chiral amino acids and vaterite boundary surfaces, the development of an accurate atomic model of vaterite is crucial. However, the structure of vaterite has been a subject of debate in the literature for 50 years, with no single definitive structure agreed upon[16]. With this uncertainty of vaterite crystal structure in mind, we explored two potential mechanisms—with each mechanism based on a different proposed crystal structure of vaterite—to explain the formation of chiral vaterite toroids induced by chiral acidic amino acids. In mechanism 'A', chiral amino acids adsorb onto an achiral symmetric vaterite surface, producing a chiral interface that then directs toroid growth in a chiral manner. In mechanism 'B', chiral amino acids preferentially nucleate vaterite nanohexagons, which are intrinsically chiral (the crystal structure belongs to a chiral space group) and each vaterite enantiomer then produces toroids with their respective chiral structure.

**Chiral amino-acid adsorption on symmetric vaterite.** This mechanism 'A' requires a vaterite model surface that both agrees with the experimental *d*-spacings and is symmetric (achiral). We explored several vaterite structural models and surfaces; as described in the Supplementary Discussion, the (100) surface of a net-neutrally terminated modified vaterite structure from Kamhi[28] meets both these requirements. To determine whether adsorption of L- or D-Asp onto this model surface produces a pair of structurally distinct chiral monolayers, we used RosettaSurface docking simulations[35] to identify low-energy conformations of the amino acid Asp bound to the modified Kamhi vaterite (100) model. On the modified

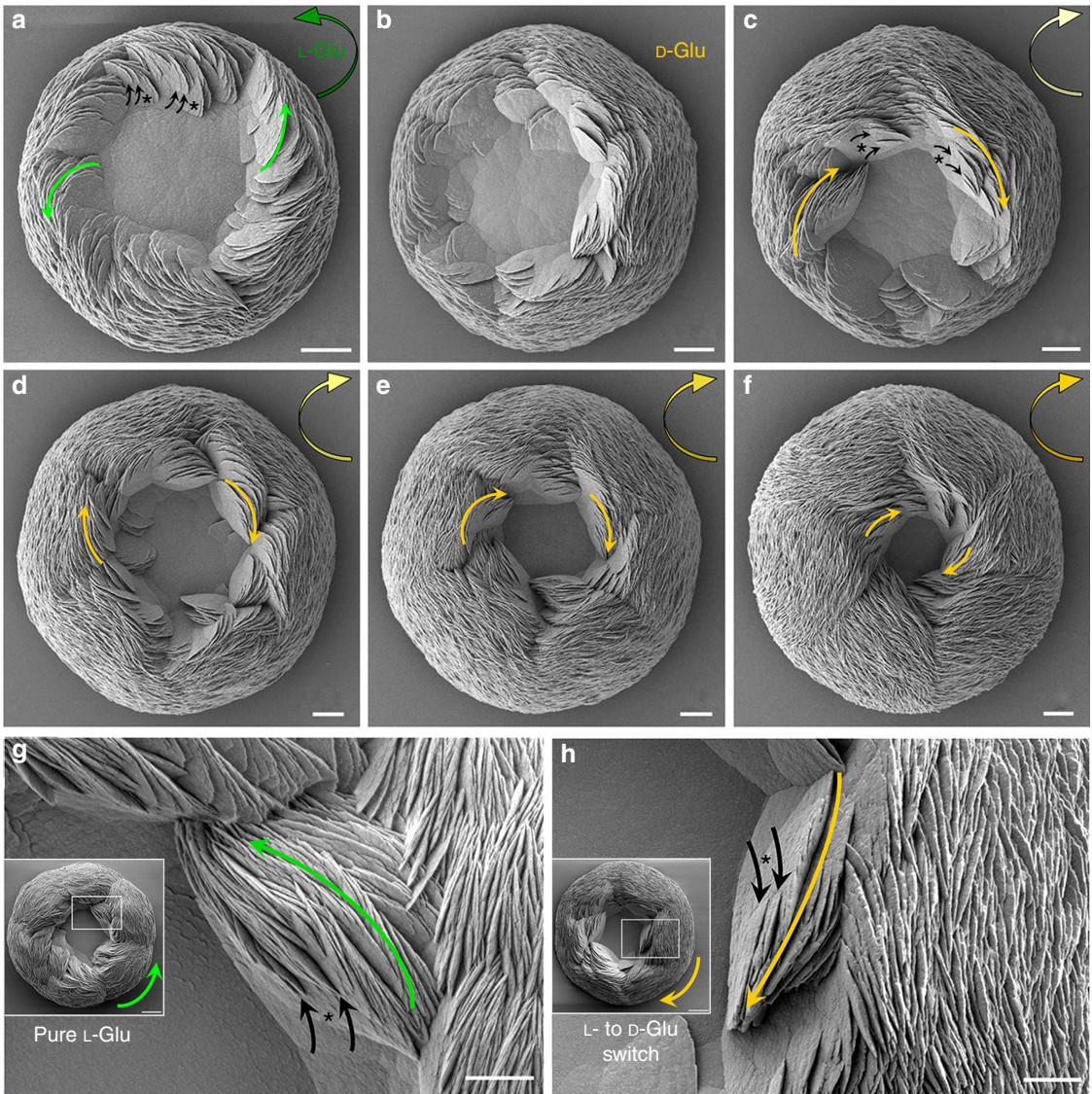

**Figure 7 | Change in toroid chirality induced by switching amino-acid enantiomer. (a)** SEM images of an early vaterite toroid with counterclockwise spiralling (green arrow) induced by L-Glu after 8 h of growth. Oriented nascent small platelets (small black arrows) branch off from existing larger platelets (asterisks) and have a counterclockwise growth direction (small green arrows in **a**) that is switched to the clockwise growth direction (small yellow arrows) by replacing L-Glu with D-Glu in the growth solution (**b–f**; 2, 3, 4, 5 and 6 days, respectively). At the transition point, mostly platelets without oriented growth direction are observed at 2 days surrounding the central region of the toroid (**b**). (**g**) High-magnification SEM images showing the oriented branching of nascent small vaterite platelets (black arrows) arising from larger existing platelets (asterisks), having counterclockwise growth direction (long green arrow) leading to the mature, chiral toroids having the counterclockwise growth direction (short green arrow, inset) for pure L-Glu. (**h**) Conversely, the clockwise growth direction (long yellow arrow) causes the mature, chiral toroids to have the clockwise growth direction (short yellow arrow, inset) after replacing L-Glu by D-Glu. Scale bars, 8 μm (**a–f**), and 10 μm for inset and 4 μm (**g,h**).

Kamhi vaterite (100) plane, enantiomers of chiral Asp recognized and discriminated between the 'left' and 'right' sides of the symmetric (100) site, with stereochemical matching occurring via a non-colinear 'three-point binding mode'[34] (Fig. 9a). Using Asp as an example, as shown in Fig. 9a, the two carboxyl groups of Asp each occupy an interstitial site between two adjacent calcium atoms of the symmetric (100) site. The bonds between the two carboxyl groups and these four calcium atoms fix the orientation of the amino acid on the mineral surface to constrain the binding geometry. In the case of L-Asp, a third bond (a hydrogen bond) readily forms between the positive amino group and the carbonate of the left side of the symmetric site as a consequence of the fortuitously matching configuration. However, no such configurational match is found on right side

of the symmetric (100; Fig. 9a). The minimized configuration of L-Asp binding the carbonate group of the left side of the symmetric (100) surface results in a calculated binding energy 0.64 kcal mol$^{-1}$ higher than that calculated for L-Asp binding to the right side, thus demonstrating an energetic binding preference of an enantiomer to either the left or right side of the symmetric surface. Conversely, D-Asp can form the third hydrogen bond with the right side of the symmetric (100) plane, resulting in an opposite orientation for this D-enantiomer (Fig. 9a). Note that even smaller differences in adsorption energy from what we have observed have been shown to be sufficient for chiral selectivity of chiral cycloalkanes on chiral Pt surfaces[36], propylene oxide on the chiral Cu (643) surface[37,38] and 3-methyl-cyclohexanone on the chiral Cu (643) surface[38].

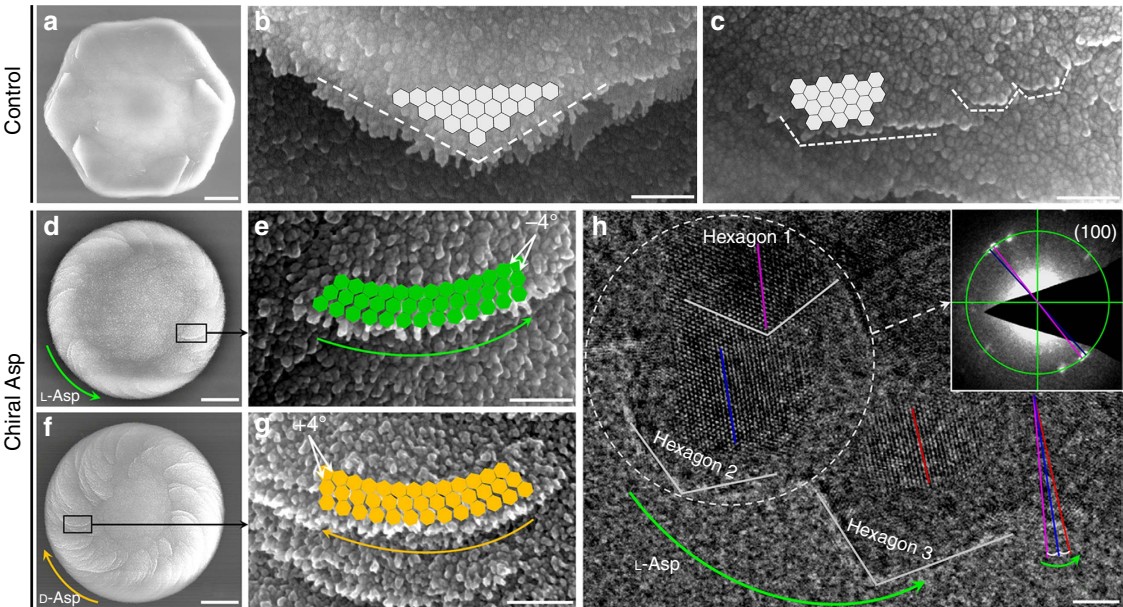

**Figure 8 | Curved-edge vaterite platelets form by tilting of vaterite nanohexagons.** SEM images without added amino acid (control) showing hexagonal symmetric vaterite at low (**a**) and high magnification (**b,c**) with straight-edge growth of platelets (white dashed lines), with matched and aligned subunit nanostructured growth (light grey hexagons), rather than the rounded/curved-edge platelet growth seen in the presence of chiral acidic amino acids. (**d–g**) Low- and high-magnification SEM images of chiral toroid platelets showing tilting (by approximately $-4°$ or $+4°$) of nanoparticle (nanohexagon) growth induced by selective chiral amino-acid adsorption. This chirality-inducing effect causes rounded/curved platelet edge growth having either a counterclockwise direction for L-Asp (green arrows) or a clockwise direction for D-Asp (yellow arrows). (**h**) High-resolution TEM image showing a counterclockwise (long green arrow) growing platelet edge formed in the presence of L-Asp with visible nanohexagons and their internal lattice structure. For consequential 'daughter' hexagon 2, its (100) plane (blue line) is tilted by approximately $-4°$ (in the counterclockwise direction, short green arrow) relative to the same plane (purple line) in the 'mother' hexagon 1. Another adjacent hexagon 3 in the field also shows the $-4°$ tilt (red line). This slight offset between the (100) planes of the hexagons is also confirmed by nanobeam electron diffraction in TEM (inset) from the white dashed-circle region. Scale bars, 4 µm (**a**), 300 nm (**b,c,e,g**), 2 µm (**d,f**) and 5 nm (**h**).

**Chiral amino-acid adsorption on chiral vaterite.** This mechanism 'B' requires a vaterite model surface that is intrinsically chiral and agrees with the experimental d-spacings. As described in the Supplementary Discussion, a recent model from Demichelis *et al.*[39] presents an enantiomeric pair of vaterite structures belonging to chiral space groups $P3_221$ and $P3_121$. The (110) surface of this structure is both intrinsically chiral and matches the experimental d-spacings. For chiral amino acids to preferentially nucleate one chiral crystal enantiomer over the other, we would expect chiral selectivity for binding of L- and D-Asp onto one of the vaterite enantiomers. To explore this hypothesis we used RosettaSurface to dock both L- and D-Asp onto the neutrally terminated Demichelis $P3_121$ (110) surface. Docking revealed that binding of L-Asp onto the $P3_121$ surface is favoured by 0.33 kcal mol$^{-1}$ over D-Asp (which, by symmetry, implies binding of D-Asp onto the $P3_221$ is favoured by 0.33 kcal mol$^{-1}$ over L-Asp). The difference in adsorption energy arises from the electrostatic complementarity between the $P3_121$ surface and L- and D-Asp (Fig. 9b). Both L- and D-Asp are anchored to the surface via hydrogen bonding between the amino terminus and the surface carbonates as well as by electrostatic contacts between the carboxyl terminus and one surface calcium. With these two functional groups locked in identical positions for both L- and D-Asp, the adsorption energy difference is attributable to the interaction of the charged side chain with the mineral. In the case of L-Asp, the side-chain carboxyl group is able to interact favourably with an exposed calcium ion. However, in the case of D-Asp, the side-chain carboxyl is forced to interact unfavourably with a highly coordinated calcium ion.

Given the uncertainties in current protein–mineral force fields, both mechanisms 'A' and 'B' are plausible explanations for the observed experimental morphology, although an advantage of our preferred mechanism 'A' is the slightly higher adsorption energy compared to mechanism 'B'. Using mechanism 'A' as an example, in the absence of chiral acidic amino acids, and as can be seen by the perfect alignment of nanoparticles in certain mesocrystals[40,41], symmetric nanostructured vaterite platelets form via a perfect orientation; in this case, hexagonal prism-shaped nanoparticles (that is, mother nanohexagon H1 spawning daughter nanohexagon H2) align to form only straight hexagonal platelet edges. Such a perfect oriented attachment for vaterite hexagonal crystals ideally gives rise to a generally coherent, single crystal-like diffraction pattern as shown by Hu *et al.*[26] with exact (100) face matching and without any tilting (Figs 8a–c and 9c). However, the adsorption of chiral acidic amino acids on the 'mother' nanohexagonal prism H1 breaks the perfectly oriented nucleation/attachment mechanism for the consequential 'daughter' nanohexagonal prism H2, resulting in a slight misaligned tilt of ~4° between the mother H1 and the daughter H2 nanohexagonal prisms (Figs 9c and 10); this is similar to the imperfect oriented attachment shown to occur between two adjacent nanocrystalline anatase particles[41,42]. More importantly, the mirror orientations of L- and D-enantiomers of chiral acidic amino acids adsorbed on the symmetrical vaterite (100) plane induce the opposite '−' (counterclockwise) and '+' (clockwise) directions of imperfect oriented nucleation (Figs 9 and 10). This simple model demonstrates the interaction between nanoparticles that might also involve a low-density, hydrated wedge of mineral ions and amino acid between nanoparticle surfaces. With further

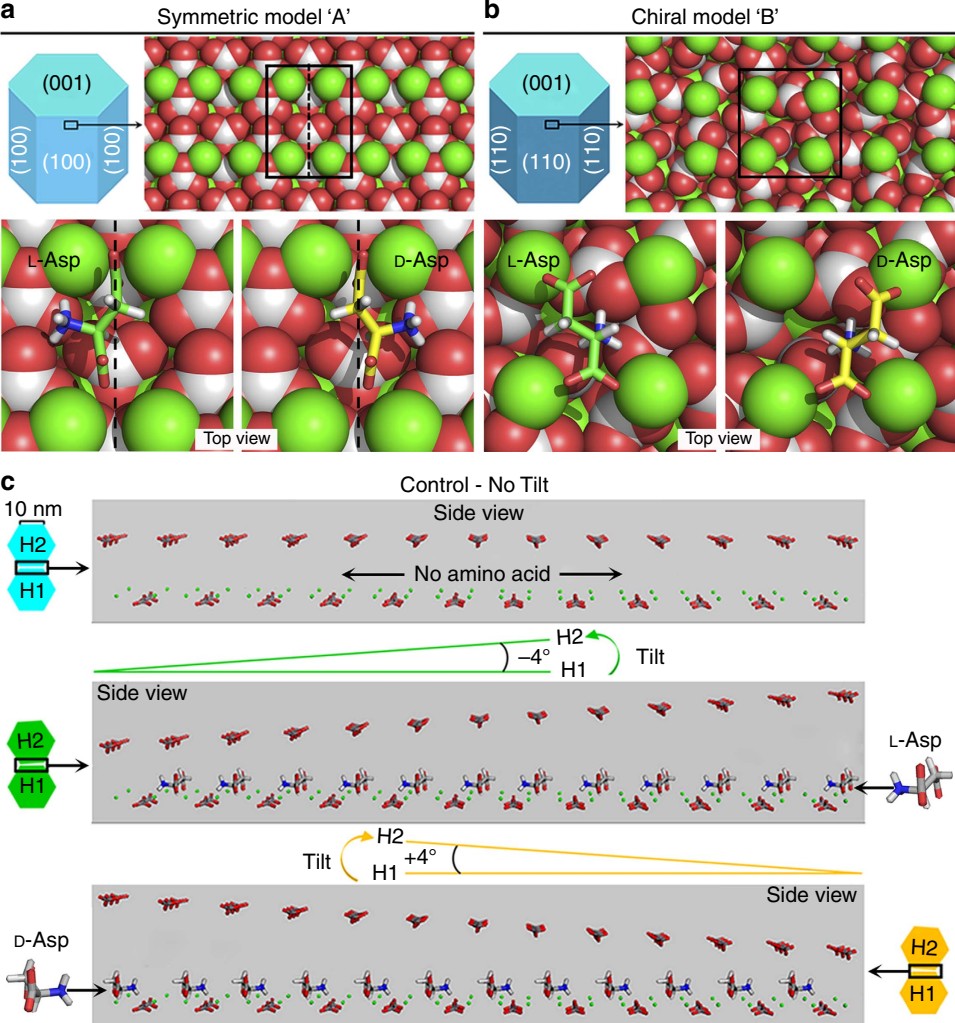

**Figure 9 | Binding geometry of chiral acidic amino acids on vaterite.** (**a**) On the hypothetical symmetric (achiral) vaterite surface of model 'A', fundamental subunit nanohexagonal vaterite prisms showing exposed symmetric binding sites (around the black dashed line within the solid rectangle) on the symmetric $P6_3/mmc$ (100) plane and the binding configurations of docked L-Asp (green) and D-Asp (yellow) involving their two carboxyl groups and one amino group. (**b**) On the hypothetical chiral vaterite surface of model 'B', configurations of docked L-Asp (green) and D-Asp (yellow) are shown on the chiral vaterite $P3_121$ (110) surface (one of a pair of enantiomeric crystal surfaces). The positions of the amino and carboxyl termini on both L- and D-Asp are identical; however, because of the chirality of the $C_\alpha$ atom, L-Asp is able to position the carboxyl group of its side chain near a more-exposed surface calcium atom, as opposed to D-Asp whose carboxyl side chain is forced to interact with a more-coordinated calcium atom. Vaterite crystal atoms: Ca, green; C, light grey; O, red. (**c**) For the symmetric (achiral) model 'A', vaterite hexagonal growth configurations showing the relationship of consequential 'daughter' nanohexagonal prism (H2) to its 'mother' nanohexagonal prism (H1; with a (100) plane in the symmetric model critical-edge size of ∼10 nm) in the absence of amino acid (no tilting, light blue), or in the presence of L- or D-Asp; the enantiomers, respectively, cause an approximately −4° tilt in the counterclockwise direction (green), or an approximately +4° tilt in the clockwise direction (yellow). These tilts arise from the symmetric chiral adsorption of each respective enantiomer with opposite orientations onto the repeating symmetric site of the mother vaterite nanohexagonal prism (100) plane. Vaterite crystal atoms: Ca, green; C, grey; O red.

amplification of this small, misaligned growth mechanism during vaterite platelet formation and extension, curved-edge platelets arise from the toroid surface, in either the counterclockwise growth direction for L-enantiomer, or in the clockwise growth direction for D-enantiomer (Figs 8d–f and 10 and Supplementary Movies 1 and 2). This platelet edge-rounding effect over larger scales is produced by a consistent, systematic replication of enantiomer-controlled misalignment (tilting) of adjacent nanohexagonal prisms.

In conclusion, this work demonstrates that complex hierarchical and chiral architectures of the vaterite polymorph of calcium carbonate can be achieved simply by the addition of chiral acidic amino acids. These findings advance our understanding of the binding and effects of acidic amino acids known to be highly abundant in biomineralization-regulating proteins, and they provide insight into nucleation mechanisms for guiding biomineral growth and for generating complex chiral hierarchical structures commonly seen in biological systems that mineralize. Our observations of interactions between chiral amino acids and symmetric surface features of calcium carbonate minerals provide insight into one possible mechanism for mineral-mediated prebiotic chiral molecular concentration and organization—all important steps in life's origins[20,34]. In addition, these findings point to the importance of nanoscale crystal–biomolecule interactions that may prove useful in the development of complex composite/hybrid functional materials.

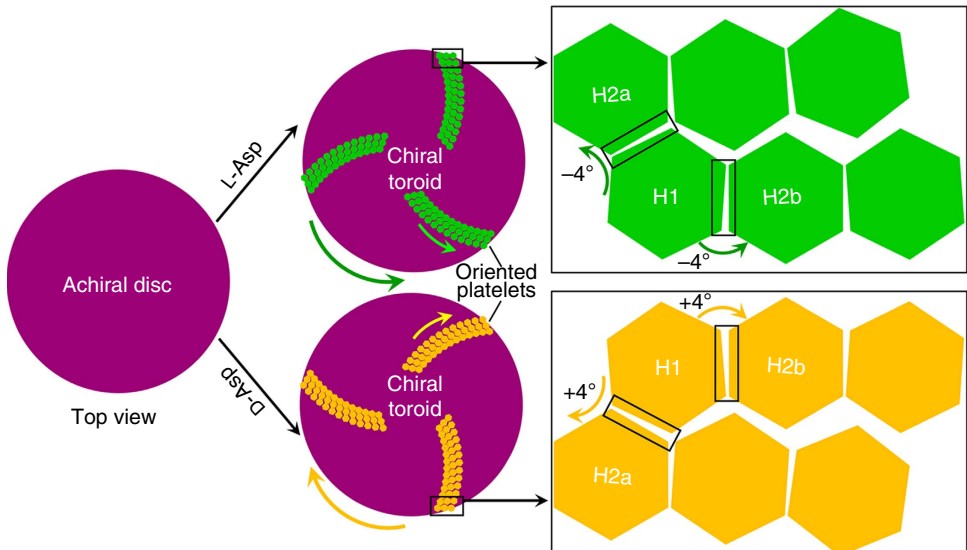

**Figure 10 | Tilting mechanism leading to the formation of chiral vaterite toroids.** Schematic summary of the formation of hierarchically organized, chiral toroids based on an approximately −4° or +4° tilting of hexagonal subunits (H1, H2a and H2b) caused by the adsorption of L- or D-enantiomer, respectively, of chiral amino acid whose amplification/replication with further growth forms thin, flat, curved-edge platelets that collectively form a chiral toroid. Note that platelets formed directly on the substrate disc are tilted rather than being parallel to the disc, by a small tilting angle of about 6° between the first-layer platelets and the substrate disc, and then between subsequent adjacent platelet layers (see also Supplementary Movie 2).

## Methods

**Calcium carbonate growth method.** In a 250 ml covered beaker, calcium carbonate was obtained by adding 50 ml of 1 mM $Na_2CO_3$ to 50 ml of a 5 mM $CaCl_2$ solution containing 2–40 mM amino acid, and the solution pH was adjusted to 10.0 ± 0.2 by stepwise addition of 1 M NaOH or HCl at 20 °C. Final concentrations of $CaCl_2$, $Na_2CO_3$ and amino acids were 2.5, 0.5 and 1–20 mM, respectively (most experiments were performed using 20 mM amino acid). In control experiments, without added amino acid, mixed calcium carbonate crystals, including achiral rhombohedral calcite and hexagonal vaterite crystals, were obtained by adding 50 ml of 1 mM $Na_2CO_3$ to 50 ml of a 5 mM $CaCl_2$ solution in a 250 ml covered beaker, with the pH adjusted to 10.0 ± 0.2 by stepwise addition of 1 M NaOH or HCl at 20 °C. Final concentrations of $CaCl_2$ and $Na_2CO_3$ were 2.5 and 0.5 mM, respectively. Solutions remained at room temperature for the duration of mineral growth, and no stirring was performed at any time. When particulate material began to appear at the bottom of the beaker (as observed by optical microscopy occurring between 12 and 24 h, and not in solution), round glass coverslips (to facilitate removal and subsequent mineral analysis) were gently dropped to the bottom of the beaker; subsequently, calcium carbonate particles then formed directly on the glass coverslips. Periodically, after different reaction times, glass coverslips with grown calcium carbonate particles were removed from the beaker and rinsed with doubly distilled water and ethanol, and allowed to dry in a vacuum desiccator for at least for 1 week at room temperature before analysis. For the detection of incorporated amino acids into mineral by micro-Raman spectroscopy and atomic force microscopy (AFM), all vaterite samples on glass coverslips were washed 3 × with 1 M NaOH for 2 min each to remove any soluble surface-bound amino acids, and then these vaterite samples on the glass coverslips were rinsed with doubly distilled water and ethanol, and allowed to dry in vacuum desiccator at least 7 days at room temperature before investigation.

**Calcium and amino-acid quantitation in chiral vaterite toroids.** The calculation of the amount of chiral acidic amino acids incorporated into the chiral vaterite toroids was determined by measuring the concentrations of calcium and chiral acidic amino acids after dissolution of the toroids as initially grown directly on glass coverslips. Calcium concentration from the dissolved vaterite was quantified using a calcium assay kit (Sekisui Diagnostics Inc, PEI, Canada). Briefly, counterclockwise ('right-handed') vaterite toroids grown in the presence of L-Asp were dissolved in 100 µl of 10% acetic acid and the calcium content was spectrophotometrically quantified using the reaction kit reagents and a microplate reader measuring at 595 nm wave length absorbance. For quantification of L-Asp incorporated into chiral toroids, an L-Amino Acid Quantification Assay (Sigma-Aldrich, QC, Canada) was performed. Vaterite grown in the presence of L-Asp was dissolved in 10% acetic acid, followed by pH adjustment to 7.00 by stepwise addition of 1 N NaOH. The L-Asp concentration was measured spectrophotometrically in a microplate reader at 560 nm. The concentrations of calcium and L-Asp were established from standard curves using biochemical assay kits and microplate reader spectrophotometric measurements.

**Sample characterization.** *Scanning electronic microscopy.* In order to improve sample conductivity, calcium carbonate grown directly on the glass coverslips was coated with a 2-nm-thick, electrically conductive layer of Pt using a Leica Microsystems EM ACE600 sputter coater (Vienna, Austria). SEM imaging was performed using an FEI Quanta 450 FE-ESEM (FEI Company) operating at a voltage of 5 kV and equipped with an Everhart-Thornley secondary electron detector.

*Transmission electron microscopy.* For TEM investigation, copper grids coated with a holey carbon film were gently dropped to the bottom of the beaker. Vaterite toroids grown directly on these grids were washed with water and ethanol, dried and analysed using a Tecnai $G^2$ F20 (FEI Company) high-resolution transmission electron microscope operating at 200 kV. For the lower-magnification images, diffraction-contrast mode was used, and for the higher-magnification lattice images, the phase-contrast mode was used. Images were taken from platelets located at the very edge of the toroids where sufficient sample thinness allowed morphological inspection and high-resolution lattice imaging.

*Atomic force microscopy.* AFM simultaneously produces both surface topography (height images or derived amplitude images) and phase images. Phase images, generated by AFM cantilever frequency shifts caused by differences between organic matrix and inorganic vaterite compositions based on their viscoelastic properties and adhesion forces, can be used to distinguish organic versus inorganic components of biomaterial samples. We obtained AFM height and phase images using a Nanoscope IIIa (Veeco, Santa Barbara, CA, USA) operating in the tapping mode with a vertical engage E-scanner and NanoScope version 5.30 software (Veeco/Bruker-AXS Inc, Madison, WI, USA), operating at room temperature in air. V-shaped tapping mode AFM probes (typical tip apex radius of ∼7 nm) with Si cantilevers having a spring constant $k = 42 \, Nm^{-1}$ (Bruker-AXS) were used. Scans were performed at rates of 0.1–2 Hz and with feedback control parameters set in the following ranges: integral gain 0.4–0.8, proportional gain 0.6–1, look-ahead gain 0–0.5 and Z limit 3.52 µm. To reduce imaging artefacts, we optimized the tip force exerted on the surface with the amplitude set-point being as high as possible.

*FIB–SEM.* We prepared ultrathin TEM lamellae of cross-sections of toroids using a Helios Nanolab 660 DualBeam (FEI Company) using conventional FIB techniques. To prevent ion beam surface damage to the toroids during the process of FIB milling, we applied a dual-cap protection layer of carbon in two steps: electron beam-induced deposition of carbon ∼100 nm thick, followed by ion beam-induced deposition of carbon ∼2 µm thick. Material surrounding the region-of-interest was removed using a 30 kV ion beam and a current ranging from 21 to 2.5 nA. Conventional *in situ* lift-out of FIB lamellae was carried out using an Easylift nanomanipulator. Lamellae were then continuously thinned from both sides using smaller beam currents (<0.79 nA). The final milling/cleaning step was carried out using a 5 kV ion beam and a current of 68 pA to obtain an electron-transparent thickness of ∼100 nm.

*Micro-Raman spectroscopy.* To measure calcium carbonate phase evolution at different growth time points, and to confirm amino-acid incorporation into the

chiral vaterite toroids, micro-Raman spectroscopy was performed using a Renishaw Via Raman microscope (Renishaw, Gloucestershire, UK) equipped with a holographic spectrometer and a Leica DM2500 M optical microscope (Leica Microsystems GmbH, Wetzlar, Germany). The excitation source was a 514.5 nm argon laser with a laser spot size of *ca.* 2 μm and an excitation power of 25 mW. The laser was focused through a ×50 objective (numerical aperture of 0.75) on a single toroid as grown on a glass coverslip. We acquired each Raman spectrum typically for 10 s, and 10 scans were accumulated for each measurement in order to minimize noise effects. Several spot analyses were taken from each selected area in order to confirm spectral reproducibility. All spectra were acquired at room temperature with a spectral resolution of $1 cm^{-1}$, and the measurement range was from 125 to $3,205 cm^{-1}$. Calibration was performed using the $520.5 cm^{-1}$ band of a silicon wafer as a standard. We performed Raman data acquisition using the Renishaw WiRE 3.4 (Windows-based Raman Environment) software. The OriginLab 6.1 software was used for spectral analysis.

*X-ray diffraction.* We measured X-ray diffraction spectra of vaterite samples (powdered and a single toroid) after growth with chiral acidic amino acids, using a D8 Discover diffractometer (Bruker-AXS) equipped with a copper X-ray tube (wavelength, 0.154056 nm) and a HI-STAR general-area detector diffraction system (GAADS; Bruker-AXS). All diffractometer components—X-ray source, sample stage, laser/video alignment/monitoring system and detector—were mounted on a vertical θ–θ goniometer. The X-ray beam was guided and collimated towards the sample by a Göbel mirror parallel optic system and a pinhole collimator. The diffractometer was operated with the sample positioned in the horizontal plane (which was kept fixed), while both X-ray source and detector moved on goniometer tracks. The angle made by the X-ray source with the sample surface in the vertical plane is noted as θ1 (θ1 track), while the angle subtended by the detector with the sample surface in the same vertical plane is noted as θ2 (θ2 track). For the single-toroid analysis, an isolated vaterite toroid grown on a glass coverslip was selected; other surrounding toroids were removed/ scraped from the coverslip with a razor blade under a dissecting microscope to eliminate any possibility that addition signal might arise from additional toroids; see Supplementary Fig. 5E). The toroid analysed was *ca.* 100 μm in diameter. The X-ray beam spot was *ca.* 500 μm in diameter, and was centred on the particle. Data were acquired as diffraction patterns (in frame format) with both the X-ray source and detector kept stationary for the time of measurement. Spectra were obtained by measuring multiple subsequent frames with a frame width of 23 degrees. Output raw data are arc segments of Debye diffraction rings bi-dimensionally recorded by the GAADS detector during measurement of a frame. Diffraction rings were converted to typical 2θ diffractograms. Measurements were run in coupled θ–θ scan mode at θ1 = θ2 and with the sample–detector distance set to 30 cm.

*Computational simulation.* We produced computational models by performing Monte Carlo-plus minimization-based docking of acidic amino acids on vaterite surfaces using the RosettaSurface algorithm[35].

*Chiral acidic amino-acid models.* L- and D-amino acids were constructed with ideal bond lengths and angles[43]. The individual amino acids were represented as zwitterions with charged amino and carboxylate termini. Both the side-chain torsion angles and torsion angles about the amino and carboxylate termini were flexible.

*Vaterite modified Kamhi (100) symmetric (achiral) model.* We constructed a vaterite (100) slab using CrystalMaker (version 9.1.2) and vaterite unit cell coordinates from Kamhi[28]. As described in both the main text and Supplementary Text, the carbonate orientation in the Kamhi model is ambiguous; therefore, the model was modified by selecting a carbonate orientation consistent with our symmetry-breaking hypothesis. The thickness of the slab (1.2 nm) was chosen to exceed the maximum interaction distance in the Rosetta energy function, and the length and width (5 × 5 nm) were chosen to be large enough to remove edge effects. Several mixed-charge terminations of the bulk crystal were selected, and preliminary docking studies were performed until a plausible symmetric binding site was identified (see Supplementary Text). During Monte Carlo-plus minimization docking, the surface atom coordinates remained fixed. Although pH-dependent hydration species play an important role in hydroxyapatite and silica surfaces[44], the presence of hydration species (such as $HCO_3^-$) in calcite is debated[45–47]. Here, to focus on the structure of the biomolecule, we fixed the bulk crystal protonation states and used a pristine termination.

*Vaterite Demichelis (110) chiral model.* We constructed a vaterite (110) slab using CrystalMaker (version 9.1.2) and vaterite unit cell coordinates from Demichelis *et al.*[39]. The slab dimensions were identical to the modified Kamhi (100) model (above), and once again a net-neutral pristine termination was selected. The vaterite model proposed by Demichelis *et al.*[39] is, in fact, a pair of enantiomeric structures belonging to chiral space groups ($P3_221$ and $P3_121$). In this study, a single crystal enantiomer ($P3_121$) was selected for docking against both L- and D-amino acids to determine chiral selectivity.

*Energy function.* We used calcium carbonate parameters of Raiteri *et al.*[48], which have been employed for molecular adsorption[49], nucleation[50] and surface morphologies[51], with the Rosetta Talaris-2013 energy function[52], which has been tested in wide contexts and includes a combination of terms for van der Waals energies, hydrogen bonds, electrostatics and solvation via an implicit-solvent Gaussian exclusion model[53]. After initial docking simulations produced low-energy models that displayed weak oriented preference, we increased the contribution of the orientation-dependent hydrogen-bonding term. The lowest-energy models

generated with this modified score function achieved high-quality hydrogen bonds that conferred the oriented surface-binding preference. Note that the implicit water model used enables direct biomolecule–surface contacts without the limitation of water-drainage timescales, with the trade-off of necessarily omitting water-mediated hydrogen bonds and local structuring.

*Simulation algorithm.* We performed rigorous sampling of individual amino acids on vaterite by selecting a random orientation of the amino acid and placing it at a random position within the surface unit cell of vaterite symmetric (100) plane of the modified Kamhi model, and chiral crystal enantiomer ($P3_121$) of the (110) plane of the Demichelis model[28,39]. Computational docking was then performed using the RosettaDock algorithm, which samples rigid-body orientation and amino-acid conformations. Finally, at the conclusion of docking, the structure was energy-minimized using a line search employing quasi-Newton minimization with the Broyden-Fletcher-Goldfarb-Shanno-Hessian update method, the size of each step was determined exactly, meaning that the step is to the true (local) minimum of the function along that line[54]. The rigid-body orientation of the amino acid relative to the vaterite surface and all torsional angles within the amino acid were flexible during minimization.

**Data availability.** The data that support the findings of this study are available from the corresponding author (M.D.M.) upon reasonable request.

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

## Acknowledgements

We greatly appreciate help from Dr David Liu, Ms Line Mongeon and Dr Qian Wu for assistance with the mineral characterization. We thank Dr Kelly Sears and Ms Xiaoyu Tian for her discussions and suggestions, Dr Richard Chromik for the use of his micro-Raman spectrometer and McGill University's Facility for Electron Microscopy Research for assisting in many ways with this work. This study was supported by a grant from the Canadian Institutes of Health Research (to M.D.M.). M.D.M. is a member of the FRQ-S Network for Oral and Bone Health Research and the McGill Centre for Bone and Periodontal Research.

## Author contributions

W.J. and M.D.M. designed the experiments and drafted the initial manuscript. W.J., D.A., V.N. and H.V. performed the experiments, and M.S.P. and J.J.G. performed the computational simulations with input from R.M.H. All authors analysed and interpreted the data, and participated in the writing and editing of the manuscript. M.D.M. provided overall supervision of the research.

## Additional information

**Competing interests:** The authors declare no competing financial interests.

