## [Peer Review File · Nature Communications]

Reviewer #1 (Remarks to the Author):

In this paper by Wenge et al. it is beautifully shown how a single amino acid which is introduced during crystallization of vaterite can control the chirality of an ensemble of vateritic aggregates. Further it is shown that changing the chirality of the amino acid in a corresponding manner changes the chirality of the ensemble while if a racemic mixture of a specific amino acid is used, no chirality is observed in the mineral.

Not only do the authors present first class experimental and characterization data, but moreover they have studied the possible mechanism via modeling. The model proposed is well explained and though it is not definite it is indeed a plausible explanation and fits well with the experimental data.

I have no doubt that this paper will have broad impact on the field of biomineralization, bio-inspired synthesis and chemistry and materials science in general. I have seen this work presented in the GRC on biomineralization and am sorry it has not been published long ago as it deserves. I strongly recommend accepting the paper for publication as is.

If the authors wish, I would suggest citing a paper from 2012 that showed incorporation of single amino acids into CaCO₃. though this is not a demand, I believe that in this paper the amino acids are indeed getting incorporated into the lattice of vaterite.

Reviewer #2 (Remarks to the Author):

The topic of chirality and its role in mineral nucleation/growth is clearly of great significance for the area of biomineralization. There has been a lot of discussion of the specific interaction of chiral amino acids with surfaces such as the basal plane of calcite, which is readily amenable to study via AFM and other methods. Here chiral influence has been detected, such as in C. Wu et al, Cryst. Growth. & Des., 12, 2594 (2012). What is particularly different about this very interesting piece of work is that it provides a connection between amino acid surface binding (as elucidated by AFM and molecular modeling) and hierarchical structure of the assembly of nanoparticles of vaterite. To the best of my knowledge this is something that has not been shown before and makes the work significant and of broad general interest. Therefore I would support the publication of the work in this journal.

I have just a few small points for the authors to consider in finalizing the manuscript:

- The authors chose the Kamhi structure for vaterite rather than more recent ones that provide ordered supercells. In the supporting information there is a discussion how surface carbonates were modified to deal with the partial occupancy. However, it wasn't quite clear what happened in the bulk region of the slab? Also, was the final surface actually one that corresponded to one of the ordered models then? It might be as well in the main text to refer to it as a "modified Kamhi structure" otherwise those that don't read the SI might be puzzled.
- In the supplementary information the authors discuss the possibility that vaterite might have an intrinsically chiral structure, as proposed by Demichelis and co-workers. This also gives discrimination in the surface binding for the chiral amino acids without having to make somewhat arbitrary reconstructions of the surface (i.e. without justifying the reconstruction in terms of thermodynamics). In the end the first model is chosen because it gives the larger energy difference. However, given the uncertainty in force field parameters this doesn't seem to be reasonable grounds for choosing this model over the second one. I think it would be better to highlight two possible models in the main

paper (one where surface chirality is created by local modification and one where it is intrinsic to the structure) and then say that both are consistent with the results. Showing consistency regardless of the precise structural details equates with a more robust conclusion and this is too important to bury in the SI.

- Supplementary figure 11 - this connects to the previous point in part. Which way is the surface oriented in this figure? Where is the N & H as per the caption? Looking at this surface it doesn't look like a nice stable configuration to expose to solution, though without knowing the orientation it is hard to tell.

- For the calculations a combination of the force field of Raiteri et al and the Rosetta Talaris-2013 was used. Exactly how were they combined? If using the explicit force field then there is explicit electrostatics & so the use of finite slab models might be more problematic since there could be a net dipole or other moment. Was this an issue?

- "energy-minimized using an exact line search with the Broyden-Fletcher-Goldfarb-Shanno update method". Just a couple of minor things here. I presume the authors mean a Newton-Raphson minimization, since BFGS is a Hessian updating algorithm, and I don't know what makes for an "exact" line search. Most line searches are approximate in that they assume quadratic behavior (which may not be quite correct) and have a numerical convergence tolerance.

- Supplementary table 1: This says "Demichelis et al. (2014)", but there is no paper in the references with this year.

Reviewer #3 (Remarks to the Author):

The article by Jiang et al describes a fascinating observation – that chiral amino acids can induce chirality in the morphology of vaterite crystallites. This is a surprising observation that I would never have predicted. However, the data are completely convincing, and are supported by modelling studies to help explain this observation. While it is very difficult to ever completely prove the origin of such an event – which is defined by molecular scale interactions between growing crystals and organic additives – the authors have made every effort to investigate the mechanism using both experiment and modelling.

The topic of chirality in biology is one that attracts enormous attention, and I expect this paper to be of interest to a wide range of readers (not only those who work in crystallization). The work is also very thorough and beautifully presented. I therefore fully support publication in Nature Communications.

(1) With Nature Communications the authors have the possibility of publishing a larger number of Figures. I would therefore strongly suggest that they take advantage of this opportunity and consider moving some of the Supplementary Figs to the main paper (as only a relatively small proportion of readers will download this). For example, I think readers would benefit from seeing Figure S3, which really emphasizes the activities of the chiral additives. Fig S2 makes the reproducibility of the results very clear, and I really like Fig 5 G, H and I.

(2) I cannot find any reference to Figs S11, S12 and S13 in the main paper. Description of the effect of pH (to which Fig 12 relates) is interesting and would strengthen the paper. I would therefore suggest that the authors add this to the main paper.

(3) I find it really, really surprising that crystallization is so slow in this system. On p5 it says that the flat disc shown in Fig2a1 is amorphous (where this corresponds to a 2h reaction time). Raman spectroscopy isn't that sensitive to crystalline material in the presence of a majority amorphous phase. XRD would be much better at following the emergence of crystalline material. Did the authors record any powder XRD data of the CaCO₃ precipitated at different time-points in the reaction? (collecting the particles from the entire reaction solution as a powder?)

Another convincing way of demonstrating a structure is amorphous (and not that the weak signal is due to less material being present) is to induce crystallization (eg by heating). As it is, the signal may also get stronger with time due to increase in size of the particles.

(4) This is not the first paper to precipitate CaCO_3 in the presence of amino acids, and yet I have never seen structures like this before. It would be useful to provide a brief review/ discussion of previous work on this topic, and to highlight what is different with the method used here.

Response to Reviewer #1

[Reviewer's Comments: In this paper by Wenge et al. it is beautifully shown how a single amino acids which is introduced during crystallization of vaterite can control the chirality of an ensemble of vateritic aggregates. Further it is shown that changing the chirality of the amino acid in a corresponding manner changes the chirality of the ensemble while if a racemic mixture of a specific amino acid is used, no chirality is observed in the mineral. Not only do the authors present first class experimental and characterization data, but moreover they have studied the possible mechanism via modeling. The model proposed is well explained and though it is not definite it is indeed a plausible explanation and fits well with the experimental data.

I have no doubt that this paper will have broad impact on the field of biomineralization, bio-inspired synthesis and chemistry and materials science in general. I have seen this work presented in the GRC on biomineralization and am sorry it has not been published long ago as it deserves.

I strongly recommend accepting the paper for publication as is.

If the authors wish, I would suggest citing a paper from 2012 that showed incorporation of single amino acids into CaCO₃. Though this is not a demand, I believe that in this paper the amino acids are indeed getting incorporated into the lattice of vaterite.]

Authors' Reply. We thank the reviewer for appreciating the novel value of our contribution, and all the positive comments, and we have added to the paper the relevant reference mentioned by the reviewer [Borukhin, S. *et al.* Screening the incorporation of amino acids into an inorganic crystalline host: the case of calcite. *Adv. Func. Mater.***22**, 4216–4224 (2012)] listed as new Ref # 31 in our revised main text.

Response to Reviewer #2

[Reviewer 2 General Comments: The topic of chirality and its role in mineral nucleation/growth is clearly of great significance for the area of biomineralization. There has been a lot of discussion of the specific interaction of chiral amino acids with surfaces such as the basal plane of calcite, which is readily amenable to study via AFM and other methods. Here chiral influence has been detected, such as in C. Wu et al, Cryst. Growth. & Des., 12, 2594 (2012). What is particularly different about this very interesting piece of work is that it provides a connection between amino acid surface binding (as elucidated by AFM and molecular modeling) and hierarchical structure of the assembly of nanoparticles of vaterite. To the best of my knowledge this is something that has not been shown before and makes the work significant and of broad general interest. Therefore I would support the publication of the work in this journal.]

Authors' Reply. Thank you for the positive comments on our work.

[Reviewer 2 Comment 1: The authors chose the Kamhi structure for vaterite rather than more recent ones that provide ordered supercells. In the supporting information there is a

discussion how surface carbonates were modified to deal with the partial occupancy. However, it wasn't quite clear what happened in the bulk region of the slab? Also, was the final surface actually one that corresponded to one of the ordered models then? It might be as well in the main text to refer to it as a "modified Kamhi structure" otherwise those that don't read the SI might be puzzled.]

Authors' Reply. We chose to use the Kamhi structure instead of a more recent one because it is both consistent with the experimental d -spacings and with our symmetry-breaking hypothesis. The final surface does not correspond to one of the ordered models, but rather it contains a mirror symmetry plane orthogonal to the (100) surface; such a plane is not found in any of the ordered models. Several of the more recent structures (including the Demichelis structure mentioned by the reviewer) are also consistent with the observed d -spacings; however, they do not contain a mirror plane orthogonal to the surface that is implicated by the d -spacings. The presence of such a mirror plane is necessary for consistency with our symmetry-breaking hypothesis. For simplicity, the carbonate orientation chosen for the surface carbonates was also used in the bulk region of the slab. Because electrostatic forces in our scoring function are truncated at 5.5 Å, changes in the carbonate orientation in the bulk region have minimal impact on amino acid binding energies at the surface. We agree that the "modified Kamhi structure" is more appropriate and we have edited the main text to reflect this.

[Reviewer 2 Comment 2: In the supplementary information the authors discuss the possibility that vaterite might have an intrinsically chiral structure, as proposed by Demichelis and co-workers. This also gives discrimination in the surface binding for the chiral amino acids without having to make somewhat arbitrary reconstructions of the surface (i.e. without justifying the reconstruction in terms of thermodynamics). In the end the first model is chosen because it gives the larger energy difference. However, given the uncertainty in force field parameters this doesn't seem to be reasonable grounds for choosing this model over the second one. I think it would be better to highlight two possible models in the main paper (one where surface chirality is created by local modification and one where it is intrinsic to the structure) and then say that both are consistent with the results.]

Authors' Reply. This is a very reasonable point, and we are in agreement that highlighting both models in the main text would present a more complete picture of the modeling work (please see our new Fig. 9), and thus we have done so, thank you. Additionally, presenting both models side by side (and equally consistent with our experiments) will be more likely to start conversations in the scientific community regarding which mechanism (surface chirality through local modification vs chirality intrinsic to structure) is more plausible. We have thus moved the Demichelis model to the main text as requested, where it is presented in terms of both a description of this model, and a figure.

[Reviewer 2 Comment 3: Supplementary figure 11 - this connects to the previous point in part. Which way is the surface oriented in this figure? Where is the N & H as per the caption? Looking at this surface it doesn't look like a nice stable configuration to expose to solution, though without knowing the orientation it is hard to tell.]

Authors' Reply. In this figure the reader is looking vertically down at the (100) surface of the modified Kamhi structure. The caption incorrectly labeled amino acid atoms N and H that are not present, this has now been corrected - sorry. Also, for the two models, panels A and B in Figure 9 and Supp. Figure 10 (previously Supp. Figure 11) are now presented in the same format (and viewed from the same direction). These images include atoms beyond those in the top-most layer and uses spheres to more easily visualize which atoms are exposed.

[Reviewer 2 Comment 4: For the calculations a combination of the force field of Raiteri et al and the Rosetta Talaris-2013 was used. Exactly how were they combined? If using the explicit force field then there is explicit electrostatics & so the use of finite slab models might be more problematic since there could be a net dipole or other moment. Was this an issue?]

Authors' Reply. Mineral partial charges (calcium, carbonate carbon and oxygen) from the Raiteri *et al.* force field were combined with amino acid atom partial charges from the Talaris-2013 scoring function. Electrostatic interactions between all partial charges in the RosettaSurface algorithm are calculated using a distance-dependent dielectric model (Warshel, A.; Russell, S. T. Calculations of Electrostatic Interactions in Biological Systems and in Solutions. *Q. Rev. Biophys.* **2009**, *17*, 283). Because we are keeping the positions of all vaterite atoms fixed during our simulations, and we are using implicit solvent, there was no need to use the force field parameters for the Buckingham potentials between calcium, carbonate, and water atoms prescribed in the Raiteri *et al.* force field. Van der Waals forces between amino acid atoms and vaterite atoms were modeled using the Lennard Jones 12-6 potential. Lennard Jones well depths and radii for amino acid atoms were assigned from the Rosetta Talaris-2013 scoring function, and well depths and radii for vaterite atoms were assigned from CHARMM.

The (100) surface of our modified Kamhi structure is a type III surface (alternating layers of cations and anions), this means that all possible terminations result in a net dipole. However, the net dipole does not present a problem in our calculations for a few reasons. First, we reconstruct the surface by removing a subset of the calciums to achieve an approximately neutral surface charge (as described in the Supplemental Information). Second, we are only accounting for interactions between the amino acid and the surface; interactions between vaterite atoms are not accounted for, and the vaterite atoms are locked in their lattice positions during simulations. Third, we are using a slab thickness that exceeds the interaction cutoff (which is smoothed to zero at the transition) in our algorithm, so the structure/termination of the bottom of the slab will not affect binding energies. Fourth, we are not performing a periodic calculation (the slab thickness and length/width are large enough to remove edge effects), so the problem of slab dipoles stacking infinitely and creating an infinite surface energy is not a concern.

[Reviewer 2 Comment 5: "energy-minimized using an exact line search with the Broyden-Fletcher-Goldfarb-Shanno update method". Just a couple of minor things here. I presume the authors mean a Newton-Raphson minimization, since BFGS is a Hessian updating algorithm, and I don't know what makes for an "exact" line search. Most line searches are approximate in that they assume quadratic behavior (which may not be quite correct) and have a numerical convergence tolerance.]

Authors' Reply. Quasi-Newton minimization using the Broyden-Fletcher-Goldfarb-Shanno Hessian update method would be most accurate (and has been updated in the text). The Hessian matrix is approximated and updated after each step along the search direction. The term “exact” is used in the literature (for instance, *IMA J Numer Anal* 1985, 5, 121-124) to contrast with “inexact” line searches. An “exact” line search determines the step size that will lead to the true (local) minimum of the function along a particular line. However, in order to find the “exact” step size more function evaluations are needed with each step. In an “inexact” line search, the step size along a line need only improve the energy by a certain amount and make the gradient a certain amount flatter. In an “inexact” search, fewer function evaluations are needed with each step because the “exact” minimum is not being determined; however, the convergence behavior of both methods is often similar because the search direction is an approximation to begin with. We have updated the main text with additional detail on the minimization routine.

[Reviewer 2 Comment 6: Supplementary table 1: This says "Demichelis et al. (2014)", but there is no paper in the references with this year.]

Authors' Reply. Thank you - the appropriate reference has been added.

Response to Reviewer #3

[Reviewer 3 General Comments: The article by Jiang et al describes a fascinating observation – that chiral amino acids can induce chirality in the morphology of vaterite crystallites. This is a surprising observation that I would never have predicted. However, the data are completely convincing, and are supported by modelling studies to help explain this observation. While it is very difficult to ever completely prove the origin of such an event – which is defined by molecular scale interactions between growing crystals and organic additives – the authors have made every effort to investigate the mechanism using both experiment and modelling.

The topic of chirality in biology is one that attracts enormous attention, and I expect this paper to be of interest to a wide range of readers (not only those who work in crystallization). The work is also very thorough and beautifully presented. I therefore fully support publication in Nature Communications.]

Authors' Reply. Thank you for these positive comments.

[Reviewer 3 Comment 1: With Nature Communications the authors have the possibility of publishing a larger number of Figures. I would therefore strongly suggest that they take advantage of this opportunity and consider moving some of the Supplementary Figs to the main paper (as only a relatively small proportion of readers will download this). For example, I think readers would benefit from seeing Figure S3, which really emphasizes the activities of the chiral additives. Fig S2 makes the reproducibility of the results very clear, and I really like Fig S5 G, H and I.]

Authors' Reply. Thank you for these suggestions, and we have thus moved the original Supplementary Fig. S3 and Supplementary Fig. S5G-I to the main text as new Fig. 4 and new Fig. 8a-c, respectively. Also, original Supplementary Fig. S2 is now main-text new Fig. 1c in our current version.

[Reviewer 3 Comment 2: I cannot find any reference to Figs S11, S12 and S13 in the main paper. Description of the effect of pH (to which Fig 12 relates) is interesting and would strengthen the paper. I would therefore suggest that the authors add this to the main paper.]

Authors' Reply. The original Supplementary Figs. S11-S13 were used exclusively to support the feasibility of our model discussed in Supplementary Information, so there were no reference to them in the original main text. In our current revised version, we have moved original Supplementary Fig. S12 to the main text as new Fig. 3.

[Reviewer 3 Comment 3: I find it really, really surprising that crystallization is so slow in this system. On p5 it says that the flat disc shown in Fig2a1 is amorphous (where this corresponds to a 2h reaction time). Raman spectroscopy isn't that sensitive to crystalline material in the presence of a majority amorphous phase. XRD would be much better at following the emergence of crystalline material. Did the authors record any powder XRD data of the CaCO₃ precipitated at different time-points in the reaction? (collecting the particles from the entire reaction solution as a powder?)

Another convincing way of demonstrating a structure is amorphous (and not that the weak signal is due to less material being present) is to induce crystallization (eg by heating). As it is, the signal may also get stronger with time due to increase in size of the particles.]

Authors' Reply. It is true that the phase transformation process from amorphous calcium carbonate (ACC) to crystallized calcite or vaterite is very fast in pure solution. However, the addition of amino acids can stabilize ACC and slow down this phase transformation process, as additionally shown by others previously [*Z. Kristallogr.* 2012, 227, 744–757; *Faraday Discuss*, 2012,159, 61-85]. Similar to this previous work, we have now provided new X-ray diffraction results (below, and new Supp. Figure 3) obtained at different points directly from intact, as-grown chiral vaterite toroids attached to the glass substrate coverslips at the bottom of the beaker (vaterite toroids form only on the glass, and not in solution. Acidic amino acids indeed stabilize ACC for 2 hours, and our new XRD data below matches our Micro-Raman data (Fig. 6) and previous studies by others.

Supplementary Figure 3 | Mineral phase evolution of chiral toroids grown in L-Asp. X-ray diffraction (XRD) pattern characteristic of early-stage calcium carbonate immature initial discs (without platelets) grown for 2 hours on a glass coverslip substrate show only a single very broad peak at about 31° of 2 theta (vertical dashed line), indicative of ACC, as also observed by Micro-Raman spectroscopy and SEM (main text Figs. 5 and 6). With further toroid growth, two sharp crystallized vaterite peaks appeared, the (002) and (100) planes (two vertical solid lines), whose intensity increased with time, while the broad ACC peak decreased. These XRD data indicate a phase transformation from the amorphous state to crystalline vaterite, which is exactly what was observed by Micro-Raman spectroscopy (main text Fig. 6).

[Reviewer 3 Comment 4: This is not the first paper to precipitate CaCO₃ in the presence of amino acids, and yet I have never seen structures like this before. It would be useful to provide a brief review/ discussion of previous work on this topic, and to highlight what is different with the method used here.]

Authors' Reply. Thank you for this suggestion. We have now added a new paragraph to discuss this first description of this chiral phenomenon that has not been previously reported. (as below, and main text page 5).

“The precipitation of CaCO₃ in the presence of amino acids, including acidic amino acids, has been widely studied due to its importance in various areas of crystallization, biomineralization and geology²²⁻²⁷. Indeed, in terms of controlling calcium carbonate polymorph formation, work similar to ours has shown induction of symmetric/achiral vaterite by acidic amino acids²³⁻²⁶. However, we show here for the first time that chiral, hierarchical vaterite toroidal suprastructures can be induced by chiral acidic amino acids. We believe that this novel observation derived from our using relatively low concentrations of calcium and carbonate ions (low supersaturation level), and longer growth times, as compared to the previous studies. To grow calcium carbonate mineral, two main methods are generally used: *i*) the fast (minutes to hours) ammonia-diffusion method which quickly results in a high carbonate ion concentration and a high pH solution (from basic ammonia ions), and *ii*) the high concentration method of solution CaCl₂ and Na₂CO₃/ NaHCO₃, both of which result in a very high supersaturation state for vaterite (at least 100 times greater than that used in our method)²²⁻²⁶. These previous, high-mineral ion concentration studies resulted in faster precipitation of vaterite (minutes to hours) compared to our slower (hours to days) process.

Consequently, under the fast-growth conditions used by others, the high concentration of calcium and carbonate ions dominated the dynamics of vaterite growth and symmetric vaterite structure was formed, as contrasted to the spiraling chiral effects we observed for vaterite by presenting acidic amino acid enantiomers under slower calcium carbonate growth conditions (Figs. 1-3, and Supplementary Fig. 1).”

Reviewer #1 (Remarks to the Author):

All I can say is: strongly recommend acceptance!

Reviewer #2 (Remarks to the Author):

The authors have address all the points raised to my satisfaction and so I'm happy to recommend acceptance essentially "as is" without further revision. I'm sure this will be a valuable paper for the community.

There are just some trivial points that can be addressed if any further changes are requested by other referees or at the proof stage:

Page 3, line 21: "the most thermodynamically stable phase of CaCO₃" - should add "at ambient conditions" since it's not universally true.

Page 4, line 4: "presence of amino acid" -> "presence of an amino acid" or "presence of amino acids"

Page 13, line 12: "which is nearly double" - though I would note the difference is only of the order of thermal energy per degree of freedom & so not particularly significant.

Page 17, lines 28/29: I'd normally say "carboxylate" rather than "carboxy", but this may be a matter of personal preference.

Page 18, line 10: "Raitieri" -> "Raiteri"

Reviewer #3 (Remarks to the Author):

The authors have addressed all comments very thoroughly, and I fully support publication of the manuscript. I look forward to seeing it published in Nature Comms.

Response to Reviewer #2

All points are responded to below, and highlighted in yellow in our new revision.

[Reviewer 2 Comment 1: Page 3, line 21: "the most thermodynamically stable phase of CaCO₃" should add "at ambient conditions" since it's not universally true.]

Authors' Reply. Thank you - "at ambient conditions" has been added.

[Reviewer 2 Comment 2: Page 4, line 4: "presence of amino acid" > "presence of an amino acid" or "presence of amino acids"]

Authors' Reply. Thank you - "presence of amino acid" has been changed to "presence of amino acids".

[Reviewer 2 Comment 3: Page 13, line 12: "which is nearly double" though I would note the difference is only of the order of thermal energy per degree of freedom & so not particularly significant.]

Authors' Reply. Thank you, and we have thus modified any precisely quantitative statement about the difference in binding energies to now read: "... although an advantage of mechanism "A" is the slightly higher adsorption energy compared to mechanism "B".

[Reviewer 2 Comment 4: Page 17, lines 28/29: I'd normally say "carboxylate" rather than "carboxy", but this may be a matter of personal preference.]

Authors' Reply. We have changed "carboxy" to "carboxylate".

[Reviewer 2 Comment 5: Page 18, line 10: "Raitieri" > "Raiteri"]

Authors' Reply. Thank you - this has been corrected.